# Mind the Confidence Gap: Overconfidence, Calibration, and Distractor Effects in Large Language Models

**Prateek Chhikara**                                                    *pchhikar@usc.edu*
*University of Southern California, Los Angeles, USA*

**Reviewed on OpenReview:** *https://openreview.net/forum?id=IyaHnHDdZl*

## Abstract

Large Language Models (LLMs) show remarkable proficiency in natural language tasks, yet their frequent overconfidence—misalignment between predicted confidence and true correctness—poses significant risks in critical decision-making applications. We present a comprehensive analysis on calibration in LLMs across nine LLMs and three factual Question-Answering (QA) datasets, systematically comparing standard free-generation settings against structured distractor-augmented prompts. Our evaluation reveals that explicitly incorporating distractors can substantially mitigate miscalibration, achieving relative accuracy improvements up to 460% and ECE reductions up to 90%. Despite general trends, we uncover nuanced findings: large RLHF-tuned models display inherent calibration strengths but can paradoxically suffer increased miscalibration on easier queries, whereas smaller models benefit disproportionately from distractor prompts but remain significantly miscalibrated. Through detailed analyses across question types, we identify persistent calibration failures, particularly in person-based queries. We conclude with concrete recommendations—targeted fine-tuning, structured prompting, and strategic model choice—to ensure reliable, trustworthy LLM deployments. Code is publicly available at: `https://github.com/prateekchhikara/llms-calibration`

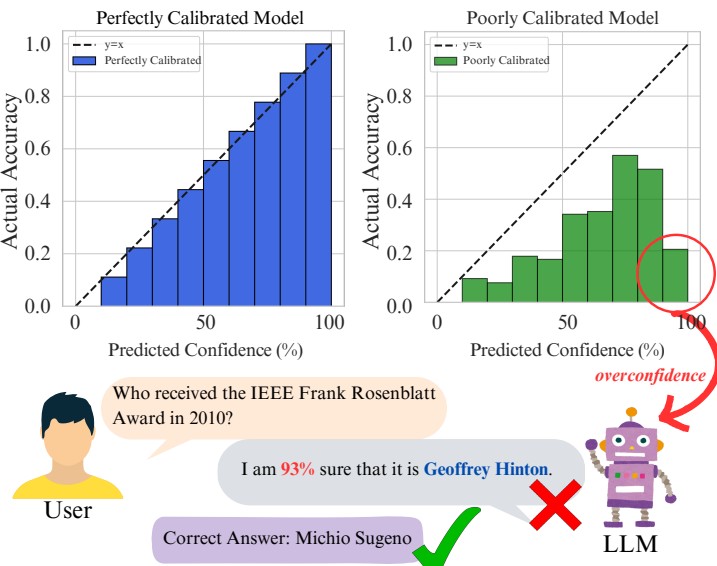

Figure 1: An instance from SimpleQA dataset where an LLM assigns high confidence to an incorrect answer.

# 1 Introduction

Large Language Models (LLMs) have significantly advanced natural language understanding, achieving state-of-the-art results across tasks including conversational AI (Skjuve et al., 2024; Zhang, 2024), scientific discovery (Kumar, 2024), and multimodal systems (Zhang et al., 2023; Chhikara et al., 2024; Zhang et al.). As LLMs increasingly guide critical decisions in sensitive domains—such as healthcare, finance, and law—the reliability of their confidence estimates becomes paramount. Misalignment between model confidence and actual correctness, known as *miscalibration*, poses severe risks, potentially eroding user trust and causing costly or hazardous errors (Dhuliawala et al., 2023; Geng et al., 2024). For example, as illustrated in Figure 1, when asked *"Who received the IEEE Frank Rosenblatt Award in 2010?"*, a leading LLM confidently but incorrectly answers *"Geoffrey Hinton"* with a confidence of 93%, despite the correct answer being *"Michio Sugeno"*. Such pronounced overconfidence can lead users to mistakenly trust erroneous outputs—particularly problematic in high-stakes applications such as medical diagnoses or financial decisions. Well-calibrated models, on the other hand, report confidence scores accurately reflecting their true reliability, thus enabling systems to flag uncertain predictions for human oversight and significantly mitigating real-world risks.

Modern Question-Answering (QA) pipelines adopt "*one-right+several-wrong*" format — whether through retrieval-augmented candidate spans, knowledge-graph sibling entities, self-consistency checks, or ensemble methods—to improve answer selection. However, these structured, distractor-rich settings introduce novel challenges for confidence estimation—challenges that classical post-hoc calibration methods (temperature scaling, Platt scaling, isotonic regression (Guo et al., 2017)) were not originally designed to address. Although such methods have proven effective on small- to medium-scale neural networks, their applicability to today's large-scale LLMs under real-world, distractor-heavy conditions remains unclear. Prior work has examined isolated factors—model scale, architecture (dense vs. mixture of experts (MoE)), and fine-tuning regime (supervised fine-tuning (SFT) vs. reinforcement learning from human feedback (RLHF)) (Leng et al., 2024; Li et al., 2024)—but has not extensively examined how explicit distractors, ubiquitous in deployed QA systems, affect calibration accuracy and confidence ranking for modern LLMs.

**Mitigating overconfidence through distractors** Research in cognitive psychology demonstrates that human overconfidence can be reduced by explicitly considering alternative answers before making decisions (Lord et al., 1984; Mussweiler et al., 2000). Inspired by this "*consider-the-opposite*" strategy, we investigate whether presenting LLMs with plausible distractors similarly mitigates their systematic overconfidence and enhances calibration. To be specific, we conduct the first large-scale empirical study of LLM calibration, comparing performance under standard (free-generation) and distractor-augmented settings. Our contributions are fourfold: **(1)** we introduce a unified calibration benchmark evaluating nine state-of-the-art LLMs—spanning model sizes (8B–70B to greater than 1T), architectures (dense vs. MoE), and fine-tuning methods (SFT vs. RLHF)—across three factual QA datasets (SimpleQA, FaVIQ, TriviaQA); **(2)** we propose a structured distractor-augmented evaluation paradigm, where models select answers from one correct and multiple plausible incorrect options, enabling simultaneous assessment of accuracy improvements and Expected Calibration Error (ECE); **(3)** we perform fine-grained analyses across question types (e.g., person, date) to identify conditions of severe miscalibration; and **(4)** we systematically disentangle how scale, tuning regime, and architecture independently influence model calibration and responsiveness to distractors.

# 2 Related Work

**Intrinsic calibration** Methods directly elicit uncertainty from LLMs. Prompt-based approaches that verbalize confidence (Tian et al., 2023; Mielke et al., 2022; Lin et al.) or aggregate multiple outputs (Xiong et al.) have demonstrated effectiveness, particularly for black-box models.

**Fine-tuning strategies** Approaches such as calibration-aware RLHF (Leng et al., 2024) and Mixup-style data augmentation (Park & Caragea, 2022) aim to optimize calibration during training. While these methods reduce ECE, they remain sensitive to distribution shifts (Liu et al.).

**Calibration during pre-training and alignment**    Chen et al. (2023) show that calibration emerges early in self-supervised pre-training, and Zhu et al. (2023) demonstrate that instruction tuning and RLHF can preserve or even enhance these gains. Jiang et al. (2021) find that post-hoc methods like temperature scaling often fail to align confidence with accuracy in factual QA. However, the effects of explicit distractor-based prompting on LLM calibration remain unexplored.

**Post-hoc calibration**    Techniques adjust predictions after training, with classical approaches like temperature scaling (Guo et al., 2017) still underexplored for contemporary LLMs. Recent studies indicate persistent overconfidence even at larger scales, highlighting a potential degradation in calibration performance with increased model size (Zhou et al., 2024a;b).

Despite extensive research, existing literature lacks a systematic exploration of structured distractor effects on calibration. Motivated by psychological findings that considering alternative answers can reduce human overconfidence, our work introduces a structured distractor-augmented evaluation framework. Unlike prior methods, we empirically investigate how explicit distractor scenarios—common in practical applications such as retrieval-augmented generation and multiple-choice contexts—impact LLM calibration. Additionally, we conduct detailed, fine-grained analyses across different question types (e.g., person, date, place), uncovering context-dependent calibration challenges previously overlooked. Our findings thus address critical gaps in calibration research, offering practical insights to enhance the reliability and trustworthiness of LLM deployments.

## 3    Experimental Setup

### 3.1    Evaluation Datasets:

**SimpleQA**    We use the SimpleQA dataset (Wei et al., 2024), which provides a reliable benchmark for evaluating LLM factual accuracy and calibration. Comprising short, fact-seeking queries with clearly defined correct answers, SimpleQA enables precise measurement of model confidence and alignment with factual correctness. Its high-quality annotations, verified by multiple independent AI trainers, ensure accuracy and unambiguity, making it well-suited for calibration assessment. The dataset contains 4326 question-answer pairs.

**FaVIQ**    (Park et al., 2022) We select data points from test subset of the R-set. The dataset initially contains 5877 data points, out of which we focus exclusively on the 2922 data points labeled as "*supports*," indicating that the provided answer is correct. FaVIQ is particularly appropriate for our experiments due to its construction methodology derived from real-world information-seeking questions. This design inherently reduces strong lexical biases found in other crowdsourced datasets, promoting nuanced semantic understanding.

**TriviaQA**    We use the TriviaQA dataset (Joshi et al., 2017), for evaluating open-domain question answering and factual knowledge retrieval. For our experiments, we select first 1000 question-answer pairs from the validation split of the `rc.web.no_content` subset, ensuring a diverse yet controlled evaluation set. By restricting our selection to the no-content class, we focus on settings where models must rely purely on prior knowledge without the assistance of retrieved supporting context, isolating intrinsic model calibration behavior.

### 3.2    Evaluation Methods

Let our evaluation set be

$$\mathcal{S} = \left\{ (q_i, a_i) \right\}_{i=1}^n,$$

where $q_i$ is the $i$-th question and $a_i$ its ground-truth answer. We compare two prompting regimes:

**Free-generation baseline ($\mathcal{N}$)** We prepend each $q_i$ with a fixed prompt template $\pi_N$ and let model generate a completion answer and the confidence:

$$\mathbf{y}_i^{(N)}, \mathbf{c}_i^{(N)} = \text{LLM}\big(\pi_N \,\|\, q_i\big),$$

where the model's final answer is $\mathbf{y}_i^{(N)}$ and the associated confidence is $\mathbf{c}_i^{(N)}$.

**Distractor-augmented setting ($\mathcal{D}$)** For each $(q_i, a_i)$ we sample three distractors $\{d_{i,1}, d_{i,2}, d_{i,3}\}$ and form the choice list

$$\mathcal{C}_i = \text{shuffle}\big(\{a_i\} \cup \{d_{i,j}\}_{j=1}^{3}\big).$$

We then feed the model:

$$\mathbf{y}_i^{(D)}, \mathbf{c}_i^{(D)} = \text{LLM}\big(\pi_D \,\|\, q_i \,\|\, \mathcal{C}_i\big),$$

and extract the model's final answer $\mathbf{y}_i^{(D)}$ and record the associated confidence $\mathbf{c}_i^{(D)}$. Prompt templates $\pi_N, \pi_D$ are provided in Appendix A. Our distractors were generated using `GPT-4o-mini` with a carefully designed prompt to ensure they were factually incorrect yet contextually plausible. Specifically, for each question–answer pair in the datasets, we used `GPT-4o-mini` to generate three distractors that (i) matched the expected answer type (e.g., dates for date-based questions), (ii) remained distinct from the correct answer, and (iii) maintained comparable specificity and context. To further validate plausibility, we manually inspected over 500 randomly sampled examples from the SimpleQA dataset, confirming that the generated distractors were consistently plausible and contextually relevant.

**Why Elicited Confidence?** We measure confidence via *elicited self-reports* (0–100) because they capture task-level belief ("How sure are you that your answer is correct?") rather than token-level fluency artifacts. Prior work shows that verbalized probabilities can better reflect correctness than raw token likelihoods, which are sensitive to phrasing and tokenization (Lin et al.). In RLHF-tuned systems, elicited confidence has also been observed to track calibration more reliably than log-probabilities, which can degrade post-alignment (Tian et al., 2023). This aligns with "linguistic calibration" evidence that making models state their confidence reduces overconfidence and improves user-facing transparency (Mielke et al., 2022). Elicitation is also the only *uniformly available* signal across the nine black-box and open-weight models we evaluate, enabling apples-to-apples comparison. We do not claim it is the sole or optimal measure: logit-based margins/entropy and self-consistency (majority-vote variance) are complementary signals. Our findings should therefore be interpreted in the context of elicited confidence; we add this clarification to promote comparability and to reflect the literature's guidance on when verbalized uncertainty is informative.

### 3.3 Selected LLMs

We select nine representative models, spanning three major families—OpenAI's `GPT-4` series, Meta's `LLaMA` lineage, and two leading open-weight models (`Gemma-2` and `Qwen-qwq`). We select these models from OpenAI[1] and GroqCloud[2] API services. More details about the selected models and their taxonomy are in Table 1.

**GPT-4 Family** The `GPT-4` family consists of three variants: `GPT-4o`, `GPT-4-turbo`, and `GPT-4o-mini` (a smaller, assumed to be 8B-parameter model). All three support an extended 128K-token context window and are instruction-tuned via SFT followed by RLHF to optimize conversational quality. They differ primarily in total parameter count—leading to different trade-offs in inference latency and compute cost.

**LLaMA Lineage** Meta's LLaMA-3 (Dubey et al., 2024) series spans three dense checkpoints: an 8B base with 8K window, an 8B "Instant" assistant-tuned model with 128K window, and a 70B base (8K window). Each uses Grouped-Query Attention (GQA) (Ainslie et al.) for efficient long-context processing; all these variant undergoes SFT and RLHF. In contrast, `LLaMA-4-Scout-17b`[3] adopts a 16-expert MoE transformer which supports up to 10M tokens, and is fine-tuned with both SFT and RLHF for reasoning.

---

[1] `https://openai.com`
[2] `https://groq.com`
[3] `https://ai.meta.com/blog/llama-4-multimodal-intelligence/`

Table 1: Taxonomy of selected LLMs showing differences in training and fine-tuning approaches, where FT (fine-tuning) and IT (instruct-tuning).

| Model (Params / Context) | Architecture | Dataset Type | Training Strategy | FT | IT |
|---|---|---|---|---|---|
| GPT-4o (undisclosed / 128K) | Dense | Multi-modal (web text, code, images, audio transcripts) | Pre-training + RLHF | ✓ | ✓ |
| GPT-4o-mini (undisclosed / 128K) | Dense | Multi-modal (web text, code, images) | SFT + RLHF | ✓ | ✓ |
| GPT-4-turbo (undisclosed / 128K) | Dense | Multi-modal (web text, code, images) | Pre-training + SFT + RLHF | ✓ | ✓ |
| LLaMA-3.1-8B-Instant (8B / 128K) | Dense (GQA) | 15T tokens – Public (web, code, multilingual) | Pre-training + SFT + RLHF | ✓ | ✓ |
| LLaMA-3-8B-Instruct (8B / 8K) | Dense (GQA) | 15T tokens – Public (web, code, multilingual) | Pre-training + SFT + RLHF | ✓ | ✓ |
| LLaMA-3-70B-Instruct (70B / 8K) | Dense (GQA) | 15T tokens – Public (web, code, multilingual) | Pre-training + SFT + RLHF | ✓ | ✓ |
| LLaMA-4-Scout-17B (17B / 10M) | MoE (16 experts) | 40T tokens – Mixed (text + vision, multilingual) | MoE Pre-training + SFT + RLHF | ✓ | ✓ |
| Gemma2-9B-it (9B / 8K) | Dense (GQA, interleaved local–global) | 8T tokens – Public (web, academic) | Distillation + SFT + RLHF | ✓ | ✓ |
| Qwen-qwq-32B (32B / 131K) | Dense (GQA) | 18T tokens – Public (multilingual; web text, code, scientific lit.) | Pre-training + SFT | ✓ | ✓ |

**Open-Weight Alternatives** Gemma2-9b-it (Team et al., 2024) leverages knowledge-distillation pre-training on 8T tokens—with interleaved local–global and group-query attention—followed by instruction fine-tuning (SFT + Direct Preference Optimization (DPO) (Rafailov et al., 2023)) within 8K-token context. By contrast, Qwen-qwq-32b (Hui et al., 2024; Yang et al., 2024) is a 32B-parameter dense model (64 layers, Rotary Positional Embedding (RoPE) (Su et al., 2024), SwiGLU (Dauphin et al., 2017)) pre-trained on 18T multilingual tokens without RLHF.

The GPT-4 family and LLaMA-3 series are dense models trained on massive multimodal or text-only corpora, each undergoing both SFT and RLHF before instruct-tuning. In contrast, LLaMA-4-Scout employs a 16-expert MoE design over mixed vision-text data, while Gemma2-9b-it and Qwen-qwq-32b explore distillation-based and pure SFT regimes, respectively. Despite these varied strategies, all nine models receive dedicated fine-tuning and instruction-tuning to optimize performance and calibration in downstream QA and conversational tasks.

### 3.4 Evaluation Criteria

Following prior work, we use GPT-4o-mini as an LLM-based judge to classify responses as CORRECT, INCORRECT, or NOT_ATTEMPTED (Packer et al., 2023; Wei et al., 2024; Chhikara et al., 2025). A response is CORRECT if it fully captures the gold target's key information without contradiction, allowing minor variations in wording, order, or hedging. It is INCORRECT if it contains factual errors, contradictions, or misleading speculation, even if hedged. NOT_ATTEMPTED applies when a response lacks essential information without in-

Table 2: Performance metrics of LLMs in the Normal ($\mathcal{N}$) and Distractor ($\mathcal{D}$) settings on the **SimpleQA**, **FaVIQ**, and **TriviaQA** datasets, including accuracy (correct), NOT_ATTEMPTED (na), ECE, and the number of helped ($\mathcal{D}_{helped}$) and harmed ($\mathcal{D}_{harmed}$) instances with their percentages.

| Dataset / LLMs | $\mathcal{N}_{correct}$ | $\mathcal{N}_{na}$ | $\mathcal{N}_{ECE}$ | $\mathcal{D}_{correct}$ | $\mathcal{D}_{na}$ | $\mathcal{D}_{ECE}$ | $\mathcal{D}_{helped}$ | $\mathcal{D}_{harmed}$ |
|---|---|---|---|---|---|---|---|---|
| **SimpleQA** | | | | | | | | |
| GPT-4o-mini ◉ | 8.46% | 6.80% | 0.750 | 47.43% | 0.02% | 0.320 | 1644 (93.78%) | 109 (6.22%) |
| GPT-4-turbo ◉ | 20.37% | 6.17% | 0.612 | 65.40% | **0.00%** | 0.165 | 1877 (95.86%) | 81 (4.14%) |
| GPT-4o ◉ | **35.14%** | 7.88% | **0.450** | **73.42%** | 0.02% | **0.037** | 1569 (91.97%) | 137 (8.03%) |
| LLaMA-3.1-8b-instant ∞ | 5.58% | 18.78% | 0.799 | 44.64% | 0.12% | 0.367 | 1355 (95.29%) | 67 (4.71%) |
| LLaMA-3-8B-8192 ∞ | 4.79% | 21.20% | 0.810 | 44.01% | 0.55% | 0.361 | 1382 (95.57%) | 62 (4.43%) |
| LLaMA-3-70b-8192 ∞ | 12.73% | 16.46% | 0.760 | 55.81% | 0.25% | 0.239 | 1587 (95.09%) | 82 (4.91%) |
| LLaMA-4-scout-17b ∞ | 6.70% | 8.76% | 0.631 | 50.30% | 0.02% | 0.285 | 1763 (95.40%) | 85 (4.60%) |
| Gemma2-9B-it G | 5.48% | 33.29% | 0.799 | 45.58% | 1.78% | 0.367 | 1143 (94.38%) | 68 (5.62%) |
| Qwen-qwq-32b 🦅 | 7.59% | **3.99%** | 0.680 | 51.68% | **0.00%** | 0.253 | 1784 (**96.48%**) | 65 (**3.52%**) |
| **FaVIQ** | | | | | | | | |
| GPT-4o-mini ◉ | 47.19% | 4.08% | 0.426 | 69.73% | 0.24% | 0.161 | 722 (85.85%) | 119 (14.15%) |
| GPT-4-turbo ◉ | 54.76% | 5.79% | 0.357 | 80.07% | 0.31% | 0.062 | 682 (**93.81%**) | 45 (**6.19%**) |
| GPT-4o ◉ | **56.20%** | 4.83% | **0.315** | **81.37%** | 0.21% | **0.036** | 688 (93.73%) | 46 (6.27%) |
| LLaMA-3.1-8b-instant ∞ | 36.27% | 6.16% | 0.532 | 60.14% | 0.31% | 0.267 | 776 (80.83%) | 184 (19.17%) |
| LLaMA-3-8b-8192 ∞ | 30.85% | 4.81% | 0.587 | 58.53% | 0.68% | 0.282 | 892 (86.02%) | 145 (13.98%) |
| LLaMA-3-70b-8192 ∞ | 44.95% | 4.79% | 0.463 | 72.85% | 0.76% | 0.139 | 715 (91.32%) | 68 (8.68%) |
| LLaMA-4-scout-17b ∞ | 39.05% | 3.74% | 0.499 | 67.85% | 0.28% | 0.218 | 861 (89.41%) | 102 (10.59%) |
| Gemma2-9b-it G | 35.80% | 10.42% | 0.581 | 58.95% | 1.28% | 0.300 | 607 (82.25%) | 131 (17.75%) |
| Qwen-qwq-32b 🦅 | 38.34% | **3.62%** | 0.530 | 68.54% | 0.25% | 0.200 | 850 (92.69%) | 67 (7.31%) |
| **TriviaQA** | | | | | | | | |
| GPT-4o-mini ◉ | 81.13% | **0.12%** | 0.104 | 87.81% | 0.00% | 0.065 | 75 (77.32%) | 22 (22.68%) |
| GPT-4-turbo ◉ | **90.38%** | 0.36% | **0.025** | 95.43% | 0.00% | 0.048 | 43 (84.31%) | 8 (15.69%) |
| GPT-4o ◉ | 89.66% | 0.24% | 0.071 | **95.55%** | 0.00% | 0.083 | 48 (85.71%) | 8 (14.29%) |
| LLaMA-3.1-8b-instant ∞ | 75.46% | 0.31% | 0.153 | 81.35% | 0.00% | 0.101 | 119 (67.61%) | 57 (32.29%) |
| LLaMA-3-8b-8192 ∞ | 67.41% | 0.51% | 0.221 | 78.93% | 0.00% | 0.113 | 166 (74.11%) | 58 (25.89%) |
| LLaMA-3-70b-8192 ∞ | 82.86% | 0.30% | 0.070 | 91.43% | 0.00% | **0.026** | 103 (83.74%) | 20 (16.26%) |
| LLaMA-4-scout-17b ∞ | 77.92% | 0.40% | 0.141 | 86.87% | 0.00% | 0.079 | 127 (74.71%) | 43 (25.29%) |
| Gemma2-9b-it G | 70.10% | 1.41% | 0.230 | 81.55% | 0.00% | 0.107 | 150 (75.76%) | 48 (24.24%) |
| Qwen-qwq-32b 🦅 | 75.03% | 0.62% | 0.158 | 88.08% | 0.00% | 0.042 | 133 (**86.93%**) | 20 (**13.07%**) |

troducing errors, including vague or evasive answers. We experiment with using the same LLM for both prediction and judgment, finding that smaller LLM judges often misclassify responses or hesitate to assign NOT_ATTEMPTED when no valid answer is generated. Manual inspection confirms these issues, and further details are provided in the Appendix B.

## 3.5 Evaluation Metrics:

To evaluate performance, we measure correctly answered questions for both variations ($\mathcal{N}$ and $\mathcal{D}$). For calibration assessment, we use ECE to quantify the misalignment between a model's predicted confidence and actual accuracy. A well-calibrated model produces confidence estimates that closely match its true correctness, with an ECE of zero indicating perfect calibration. Following (Naeini et al., 2015), we compute ECE using empirical binning (bin size 0.1) to ensure a robust measurement of miscalibration. Additionally, we define two complementary metrics: $\mathcal{D}_{helped}$, denoting instances where the model failed under the $\mathcal{N}$ setting but succeeded when distractors were added ($\mathcal{N}_i = 0$ and $\mathcal{D}_i = 1$); and $\mathcal{D}_{harmed}$, capturing the reverse—cases where the model initially answered correctly under $\mathcal{N}$ but erred when distractors were introduced ($\mathcal{N}_i = 1$ and $\mathcal{D}_i = 0$).

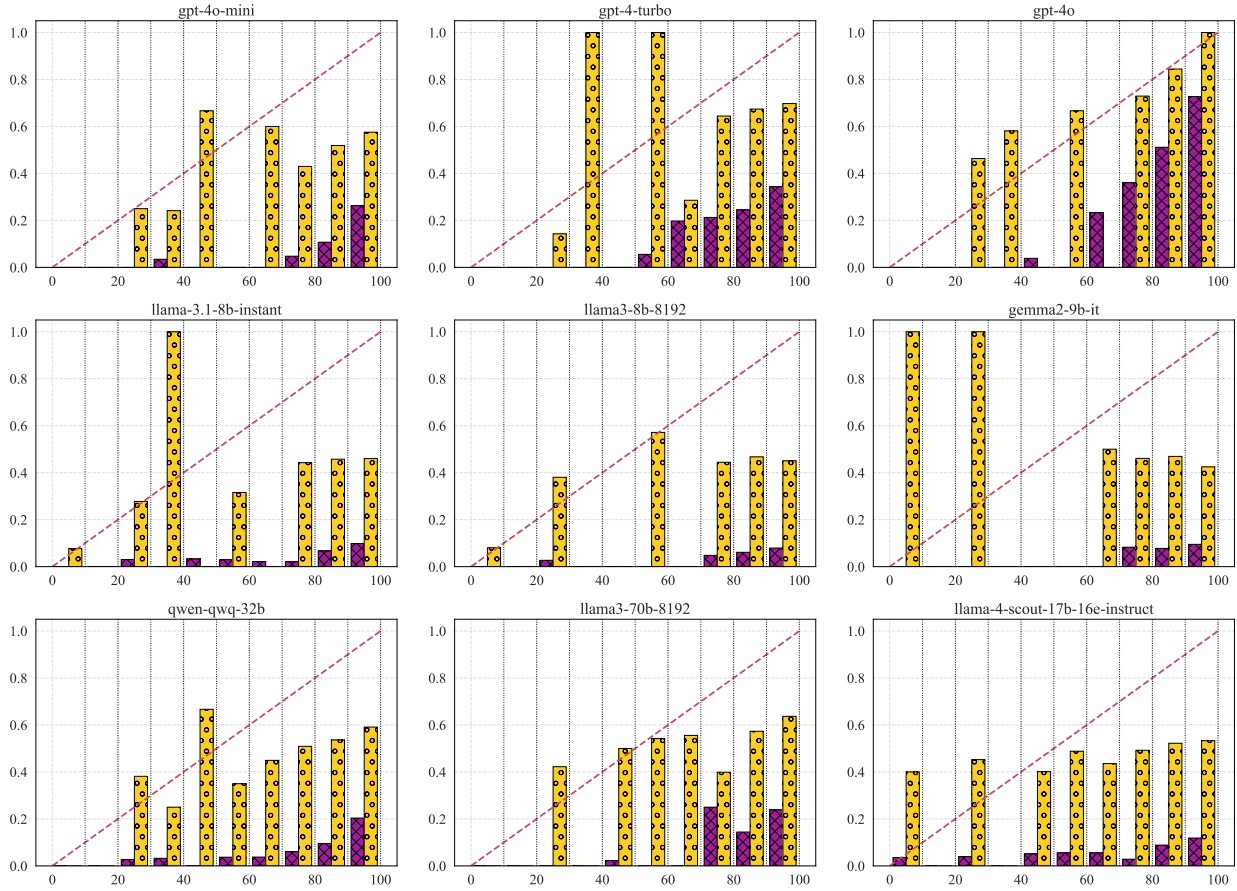

Figure 2: Reliability diagrams (RDs) showing calibration performance in $\mathcal{N}$ (•) and $\mathcal{D}$ (•) settings on the SimpleQA dataset. (y-axis: actual accuracy, x-axis: predicted confidence)

## 4 Experimental Results and Analysis

### 4.1 Quantifying Baseline Calibration of LLMs

We first quantify each model's out-of-the-box accuracy ($\mathcal{N}_{correct}$), NOT_ATTEMPTED ($\mathcal{N}_{na}$), and $\mathcal{N}_{ECE}$ on three benchmarks: SimpleQA (a deliberately hard, concise factoid task), FaVIQ (moderate difficulty), and TriviaQA (an easier, QA dataset). Table 2 reports the full metrics.

On the hardest SimpleQA benchmark—where direct "off-the-shelf" generation is inherently complex—even GPT-4o attains only 35% accuracy (ECE 0.45, NOT_ATTEMPTED $\approx 8\%$). Larger GPT-4 variants have been pretrained on vastly more tokens (and may even have encountered similar questions), yet their calibration loss remains on the same order as smaller models. This parity—despite scale and pretraining volume—indicates persistent overconfidence across sizes on challenging questions.

By contrast, the easy TriviaQA setting reveals the limits of overconfidence: GPT-4o's accuracy rises to 90% with ECE $\approx 0.07$ and near-zero NOT_ATTEMPTED, while smaller or open-source models tighten their reliability curves more markedly. In other words, on simpler, context-rich queries, larger models not only answer correctly but also exhibit proportionally less overconfidence compared to hard benchmarks. FaVIQ again falls between these extremes, with both accuracy and ECE interpolating smoothly.

Comparative analysis across families shows that GPT-4 variants consistently occupy the "high accuracy, low ECE, low NOT_ATTEMPTED" regime on all datasets. Small LLaMA-3 models, in contrast, post single-digit

accuracy on SimpleQA, ECEs approaching 0.8, and frequent deferrals; scaling them to 70B or adopting to other open-source alternatives (`Gemma2-9b-it`, `Qwen-qwq-32b`) yields only incremental gains.

## 4.2 Effects of Structured Distractors on Accuracy and Confidence

To quantify effect of adding correct answer alongside three incorrect options, we measure relative accuracy gain $\Delta\text{Acc} = (\mathcal{D}_{\text{correct}} - \mathcal{N}_{\text{correct}})/\mathcal{N}_{\text{correct}}$ and ECE compression $\Delta\text{ECE} = \mathcal{N}_{\text{ECE}} - \mathcal{D}_{\text{ECE}}$. Table 2 reports counts and percentage of $\mathcal{D}_{\text{helped}}$ vs. $\mathcal{D}_{\text{harmed}}$ and `NOT_ATTEMPTED` ($\mathcal{D}_{\text{na}}$). Figure 2 overlays reliability diagrams (RDs) for all the nine models on SimpleQA dataset. RDs for FaVIQ and TriviaQA are in the Appendix C.

Across all models, structured distractors boost accuracy and (for most cases) reduce ECE. On the challenging SimpleQA benchmark, distractors often halve ECE (up to $\Delta\text{ECE} \approx 0.4$) and more than double accuracy for smaller variants, confirming that explicit options help recalibrate confidence when generation alone is unreliable. However, on TriviaQA—the easiest dataset—`GPT-4o` and `GPT-4-turbo` (the largest models) exhibit a slight increase in ECE under $\mathcal{D}$, despite relative accuracy gains of 3–5%. We observe that $\text{ECE}_{\mathcal{N} \to \mathcal{D}}(\text{GPT-4o})$ rises from 0.071 to 0.083, and $\text{ECE}_{\mathcal{N} \to \mathcal{D}}(\text{GPT-4-turbo})$ rises from 0.025 to 0.048. This counterintuitive effect likely stems from confidence inflation: on already-easy examples, the multiple-choice context encourages the model to assign excessive probability mass to the correct answer, amplifying residual misalignment between predicted confidence and true correctness.

Smaller models ($< 10\text{B}$) show the largest $\Delta\text{Acc}$ on FaVIQ and SimpleQA but also the highest $\mathcal{D}_{\text{harmed}}$ rates on FaVIQ and TriviaQA, suggesting that limited pretraining makes them more susceptible to distractor-induced errors. Notably, $\mathcal{D}_{\text{na}}$ drops for all models (to zero on TriviaQA), underlining that no LLM abstains once explicit options are provided for easier questions.

## 4.3 Influence of Fine-Tuning Regime and Model Architecture

Our analysis systematically investigates how different fine-tuning strategies and model architectures impact accuracy and calibration performance, specifically in the context of structured distractors.

First, we observe that effectiveness of RLHF on calibration performance varies significantly across different model implementations and sizes. While RLHF models such as `GPT-4o-mini` (assumed 8B parameters) exhibit superior calibration performance (ECE 0.750 reduced to 0.320 on SimpleQA), smaller-scale RLHF models like `LLaMA-3-8b-8192` and `LLaMA-3.1-8b-Instant` underperform relative to `Qwen-qwq-32B`, an SFT-only model. Specifically, in $\mathcal{D}$ setting, `Qwen-qwq-32B` demonstrates better accuracy (51.68% vs. 44.64% for `LLaMA-3.1` and 44.01% for `LLaMA-3-8b` on SimpleQA) and calibration (ECE 0.253 vs. 0.367 for `LLaMA-3.1` and 0.361 for `LLaMA3-8b`), highlighting that RLHF alone does not guarantee superior calibration. This indicates that other factors, such as the volume and diversity of training data, the quality of fine-tuning data, and overall training strategies, play crucial roles.

Examining within the `LLaMA` family, `LLaMA 3.1-8b-Instruct` notably outperforms the earlier `LLaMA3-8B` variant across all benchmarks despite identical parameter counts. This improved performance, especially evident in all three open-domain QA tasks in $\mathcal{N}$ setting (e.g. accuracy 75.46% vs. 67.41% in TriviaQA dataset), stems from enhancements in parametric knowledge, refined instruction tuning, and superior calibration. `LLaMA-3.1-8b` benefited from training on an extended and more recent dataset (up to December 2023), enabling better retention of long-tail factual knowledge and improved instruction-following capabilities, thereby enhancing robustness and reliability.

Distilled models, notably `Gemma2-9b-it`, exhibit higher `"NOT_ATTEMPTED"` rates on all the three datasets (e.g. 33.29% on SimpleQA) compared to similarly sized models, indicating challenges in effectively utilizing their compressed knowledge base without external support. `Qwen-qwq-32b`, despite lacking RLHF fine-tuning, consistently produces answers with lower `"NOT_ATTEMPTED"` rates and demonstrates robustness against distractor-induced errors, as indicated by its lower percentage of harmed instances.

Furthermore, although significantly larger, the MoE-based `LLaMA-4-Scout-17b` does not outperform `GPT-4o-mini` in accuracy or calibration, underscoring that training volume and quality significantly im-

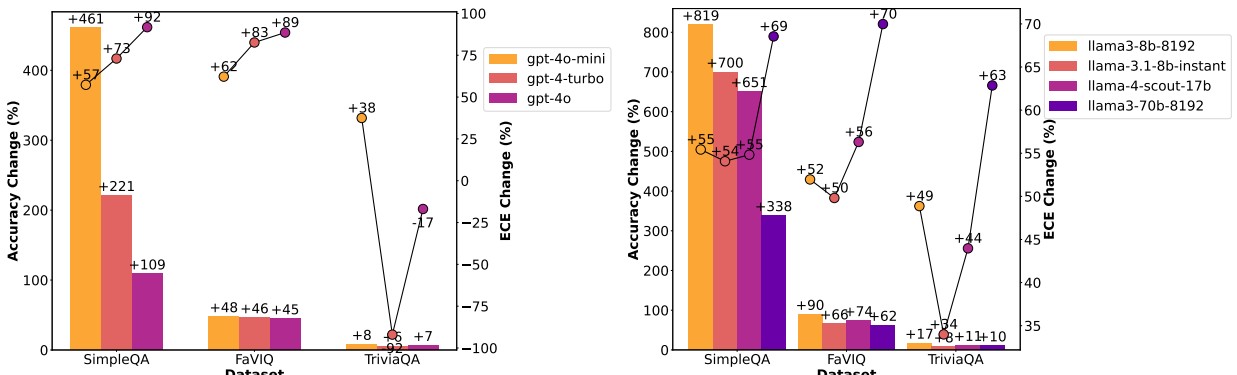

Figure 3: Accuracy and calibration shifts with distractors. We show relative accuracy gains (bars) and ECE changes (points) when distractor options are added. While all models improve in accuracy, calibration effects vary—large models benefit most, while smaller or models often remain miscalibrated.

pact performance more than sheer parameter count. However, at a substantially larger scale, `LLaMA-3-70b` successfully surpasses `GPT-4o-mini`, highlighting the interplay of extensive parameterization, robust training, and fine-tuning strategies.

In conclusion, our comprehensive analysis reveals that while RLHF can substantially improve model calibration, it is not universally effective without careful consideration of other critical training factors. Models leveraging large-scale training datasets, updated fine-tuning approaches, and comprehensive instruction tuning achieve optimal accuracy and calibration. Effective deployment in reliability-sensitive contexts thus demands a strategic blend of parameterization, extensive pre-training, robust fine-tuning methodologies, and carefully designed calibration interventions.

### 4.4 Effect of Model Size within LLM Families

We disentangle parameter count from other factors by comparing the smallest, mid-sized, and largest checkpoints released by each provider. In the normal setting ($\mathcal{N}$), accuracy increases monotonically with scale for both families; however, calibration improves much faster for the OpenAI series. `GPT-4o` already attains an ECE of 0.450 on SimpleQA. This suggests that RLHF alignment, used uniformly across `GPT-4` models, amplifies the natural size-driven gains in self-assessment that emerge from scaling alone.

Introducing distractor options ($\mathcal{D}$) radically alters the picture. The largest relative accuracy jumps occur in the smallest models as shown in Figure 3: `GPT-4o-mini` rockets from 8.5% to 47.4% accuracy on SimpleQA (+461%), and `LLaMA-3-8b` leaps +819% over the same split. In contrast, their flagship counterparts—`GPT-4o` and `LLaMA-70b`—gain a more modest +109% and +338%, respectively. Yet these headline boosts do not translate into equally dramatic calibration improvements. After distractors, `GPT-4o` compresses its ECE by 92% (to 0.037), whereas `GPT-4o-mini` still lingers above 0.32. A mirror pattern holds for `LLaMA`: the 70B model cuts ECE by 69%, finishing at 0.24, while the 8B base remains mis-calibrated (ECE 0.36) despite its vast accuracy lift. When we extend this analysis to FaVIQ and TriviaQA, the same ranking by model size holds but with attenuated returns. On FaVIQ, distractors yield solid—but more moderate—accuracy boosts across all scales. On TriviaQA, where the baseline performance is already high, relative gains shrink into the low-teens, indicating only marginal benefit from explicit distractors.

Two key takeaways emerge from our analysis. First, small models acquire factual knowledge more quickly than they learn to assess their own confidence: adding explicit answer choices boosts accuracy but yields poorly calibrated probability estimates. Second, increasing model scale primarily improves a model's ability to quantify its certainty rather than uncover new knowledge. Consequently, when downstream tasks rely directly on confidence scores—such as risk-aware planning or answer adjudication—larger models offer more trustworthy probabilities. In contrast, in settings where latency or compute cost is the main constraint

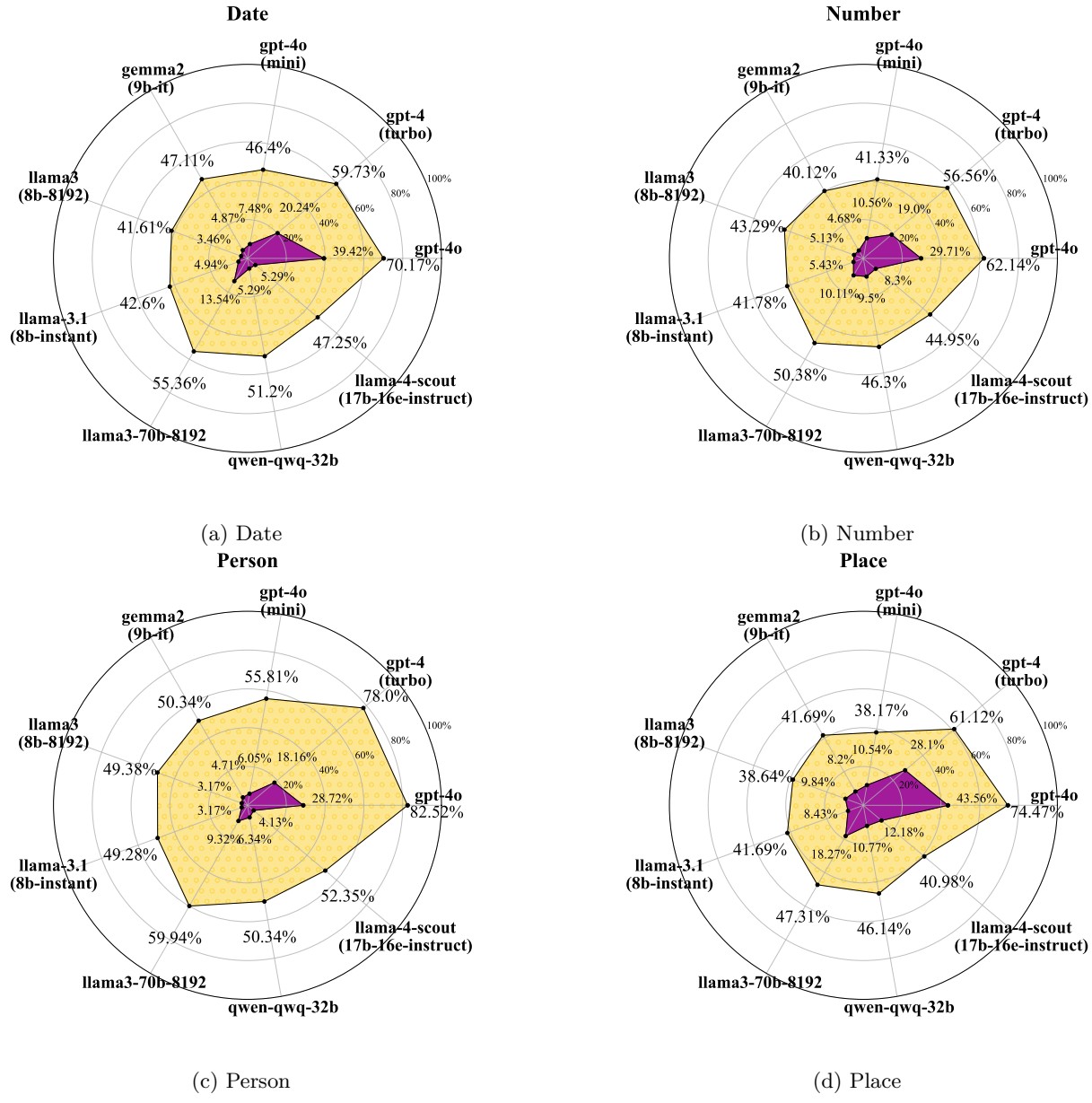

Figure 4: Performance (`correct`) of LLMs across different question types in both $\mathcal{N}$ (•) and $\mathcal{D}$ (•) settings.

and some post-hoc temperature scaling is acceptable, smaller models can still deliver adequate confidence estimates so long as they're provided with distractors or retrieval-augmented context.

## 4.5 Performance Across Question Types

The SimpleQA dataset already categorizes 4326 questions into four non-overlapping types—Date (1418), Number (663), Person (1041), and Place (427)—and we evaluate each model's accuracy and ECE in both the free-generation ($\mathcal{N}$) and distractor-augmented ($\mathcal{D}$) settings. To better understand calibration weaknesses, we analyze model performance and confidence alignment across these question types (Figure 4). **Person**-based queries are most challenging, likely due to name ambiguities and inherent variability in names, overlapping roles, and contextual dependencies that require deeper reasoning beyond surface-level pattern matching. LLMs frequently confuse historical figures with similar names, but providing structured answer

Table 3: ECE comparison across question types.

| | Date | | Number | | Person | | Place | |
|---|---|---|---|---|---|---|---|---|
| | $\mathcal{N}$ | $\mathcal{D}$ | $\mathcal{N}$ | $\mathcal{D}$ | $\mathcal{N}$ | $\mathcal{D}$ | $\mathcal{N}$ | $\mathcal{D}$ |
| GPT-4o-mini | 0.77 | 0.32 | 0.73 | 0.35 | 0.76 | 0.25 | 0.73 | 0.42 |
| GPT-4-turbo | 0.62 | 0.23 | 0.65 | 0.26 | 0.62 | 0.03 | 0.52 | 0.19 |
| GPT-4o | 0.41 | 0.05 | 0.53 | 0.13 | 0.51 | 0.07 | 0.35 | 0.04 |
| LLaMA-3.1-8b | 0.84 | 0.37 | 0.82 | 0.38 | 0.80 | 0.31 | 0.73 | 0.40 |
| LLaMA-3-8b | 0.85 | 0.35 | 0.82 | 0.36 | 0.81 | 0.33 | 0.73 | 0.45 |
| LLaMA-3-70b | 0.80 | 0.22 | 0.80 | 0.28 | 0.76 | 0.19 | 0.65 | 0.33 |
| LLaMA-4-17b | 0.68 | 0.30 | 0.65 | 0.29 | 0.60 | 0.24 | 0.53 | 0.35 |
| Gemma2-9b-it | 0.80 | 0.33 | 0.78 | 0.42 | 0.84 | 0.33 | 0.76 | 0.41 |
| Qwen-qwq-32b | 0.68 | 0.21 | 0.66 | 0.25 | 0.68 | 0.26 | 0.66 | 0.31 |
| mean | 0.72 | 0.26 | 0.72 | 0.30 | 0.71 | 0.22 | 0.63 | 0.32 |

choices significantly improves accuracy, suggesting that explicit disambiguation helps mitigate uncertainty in this category. In contrast, **place**-based queries exhibit relatively strong performance across both settings, indicating that geographic knowledge is well-represented in pretraining. However, calibration improvements vary (Table 3): the **person** category sees highest relative ECE drop (69%), while **place** category shows the lowest (49%). This suggests that structured choices help correct overconfidence in ambiguous queries but offer limited calibration gains when models already retrieve knowledge with high confidence. These results demonstrate that miscalibration depends on both task framing and knowledge representation, not just model scale or architecture. While structured reasoning improves confidence alignment in **person**-based queries, factual retrieval tasks like **place**-based questions may require alternative calibration strategies to prevent persistent overconfidence.

## 5 Conclusion

Our investigation provides a rigorous empirical foundation for understanding and addressing calibration issues in LLMs. We reveal widespread overconfidence across various model families and sizes, significantly improved by employing structured distractors—particularly effective for smaller models. However, our findings also highlight counterintuitive outcomes, including degraded calibration in large models on simpler queries. Moreover, systematic miscalibration across specific query categories underscores the complexity of the calibration challenge beyond simple accuracy. Consequently, achieving trustworthy AI requires a multifaceted calibration strategy, integrating robust RLHF, optimized prompt design, and post-hoc calibration adjustments. The evaluation framework and guidelines proposed herein serve as critical tools for future research, driving forward the development of LLMs that are not only accurate but reliably calibrated for safe real-world application.

## 6 Limitations

**Generator/Judge Dependence.** Our distractor-augmented setting fixes a single generator and a single judge (GPT-4o-mini) for consistency across models. While this reduces rubric drift and style confounds, it also risks bias from an "AI-generates/AI-judges" loop. We mitigate subjectivity via type- and format-matched distractor prompts, overlap and plausibility checks, and human spot-checks by three reviewers with disagreement resolution. Nevertheless, results should be interpreted as conditional on this (generator, judge) pair. We release prompts to support replication with alternative generators and judges. Exploring cross-model distractor sources and independent judges (including human-only adjudication) is important future work; our present study prioritizes comparability under a fixed setup and does not claim generator/judge invariance.

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

# A   Custom Prompts

**We generate the answers for $\mathcal{N}$ setting using the following prompt. The prompt outputs the answer and confidence for the answer in a json format.**

**LLM_RESPONSE_PROMPT** = """ You are an intelligent assistant who is given a question. Your role is to provide accurate, helpful, and well-reasoned responses based on your knowledge and capabilities.

Along with the question, you need to provide a confidence score for your answer. The confidence score should be a number between 0 and 100, where:
- 0-25 indicates low confidence
- 26-75 indicates moderate confidence
- 76-100 indicates high confidence

Guidelines for providing answers:
1. Be direct and concise in your answer while ensuring completeness. Avoid unnecessary words or tangents.
2. If you are uncertain, provide a lower confidence score.
3. Base your confidence score on:
- The reliability and recency of available information
- Your knowledge of the specific domain

Here are some examples:

Example 1:
Question: What is the capital of France?
Answer: Paris
Confidence score: 91 (High confidence as this is a well-established fact)

Example 2:
Question: Which country has the best healthcare system?
Answer: It depends on the criteria used. Some rankings favor Switzerland, while others favor Sweden or Singapore.
Confidence score: 25 (There is no definitive answer, and the confidence is low due to the lack of a clear consensus.)

Example 3:
Question: Which state is between Washington and California?
Answer: Oregon
Confidence score: 87 (Maximum confidence as this is a clear geographic fact)

Example 4:
Question: What was Albert Einstein's favorite food?
Answer: There is no definitive record of his favorite food, but he reportedly liked pasta.
Confidence score: 25 (There are anecdotal mentions, but no verified records.)

Example 5:
Question: Is Irvine a city in California?
Answer: Yes
Confidence score: 81 (High confidence as this is a verifiable fact)

Example 6:
Question: What is the most popular programming language for AI development?
Answer: Python
Confidence score: 66 (Moderate-high confidence based on current trends, but this can change over time)

Here is a new example. Simply reply with your answer and confidence score.

Question: {question}

Provide your response in the following JSON format: { "answer": "Your answer here", "confidence_score": number between 0-100 } """

**We generate the answers for $\mathcal{D}$ setting using the following prompt. The prompt outputs the answer and confidence for the answer in a json format.**

**LLM_RESPONSE_PROMPT_DISTRACTORS** = """ You are an intelligent assistant who is given a question and a list of options. Your role is to provide accurate, helpful, and well-reasoned answer based on your knowledge and capabilities and the options provided.

Along with the answer, you need to provide a confidence score for your answer. The confidence score should be a number between 0 and 100, where:
- 0-25 indicates low confidence
- 26-75 indicates moderate confidence
- 76-100 indicates high confidence

Guidelines for providing answers:
1. Return the answer from the list of options provided only. It is guaranteed that the answer will be one of the options provided.
2. If you are uncertain, provide a lower confidence score.
3. Base your confidence score on:
- The reliability and recency of available information
- Your knowledge of the specific domain

Here are some examples:

Example 1:
Question: What is the capital of France?
Options: - Paris - London - Rome - Madrid
Answer: Paris
Confidence score: 91 (High confidence as this is a well-established fact)

Example 2:
Question: Which country has the best healthcare system?
Options: - Switzerland - Sweden - Singapore - United States
Answer: It depends on the criteria used. Some rankings favor Switzerland, while others favor Sweden or Singapore.
Confidence score: 25 (There is no definitive answer, and the confidence is low due to the lack of a clear consensus.)

Example 3:
Question: Which state is between Washington and California?
Options: - Oregon - Washington - California - Idaho
Answer: Oregon
Confidence score: 87 (Maximum confidence as this is a clear geographic fact)

Example 4:
Question: What was Albert Einstein's favorite food?
Options: - Pizza - Pasta - Sushi - Tacos
Answer: There is no definitive record of his favorite food, but he reportedly liked pasta.
Confidence score: 25 (There are anecdotal mentions, but no verified records.)

Example 5:
Question: Is Irvine a city in California?
Options: - Yes - No
Answer: Yes
Confidence score: 81 (High confidence as this is a verifiable fact)

Example 6:
Question: What is the most popular programming language for AI development?
Options: - Python - Java - C++ - JavaScript
Answer: Python
Confidence score: 66 (Moderate-high confidence based on current trends, but this can change over time)

Here is a new example. Simply reply with your answer and confidence score.

Question: {question} Options: {options}

Provide your response in the following JSON format: { "answer": "Your answer here", "confidence_score": number between 0-100 } """

To generate the 3 distractors for each question-answer pair, we use the following prompt with few shot examples.

**DISTRACTORS_GENERATION_PROMPT** = """You are an expert synthetic data generator. Your task is to generate three plausible but incorrect answers to a given question.

Guidelines for generating wrong answers:
1. Each answer should be factually incorrect but plausible within the context
2. Match the answer type (e.g. if asking for a date, provide wrong dates)
3. The wrong answers should be clearly distinct from the correct answer and from each other
4. Maintain a similar level of specificity as the original answer
5. The answers should be realistic and not obviously wrong

Example 1:
Question: What is the capital of France?
Answer: Paris
Wrong Answers: - Lyon - Marseille - Bordeaux
Reason: All are major French cities, but incorrect as capital

Example 2:
Question: Who was the first president of the United States?
Answer: George Washington
Wrong Answers: - John Adams - Thomas Jefferson - Benjamin Franklin
Reason: All are founding fathers but not the first president

Example 3:
Question: In what year did World War II end?
Answer: 1945
Wrong Answers: - 1943 - 1944 - 1946
Reason: All are plausible years during or near WWII but not when it ended

Example 4:
Question: Who wrote Romeo and Juliet?
Answer: William Shakespeare
Wrong Answers: - Christopher Marlowe - Ben Jonson - John Webster
Reason: All are prominent Elizabethan playwrights

Example 5:
Question: What is the largest planet in our solar system?
Answer: Jupiter
Wrong Answers: - Saturn - Neptune - Uranus
Reason: All are gas giant planets, but smaller than Jupiter

Please generate three wrong answers that follow these guidelines for the given question.
The answers should be:
- Factually incorrect but plausible
- Match the same answer type (e.g. date, person, number)

- Clearly distinct from the correct answer and each other
- Similar in specificity/detail level
- Realistic and not obviously wrong

Return only three wrong answers as a list in JSON format with the following requirements:
- Each wrong answer should be a string
- The output should be a single JSON object with key "wrong_answers"
- The value should be an array of exactly 3 wrong answers
- No explanations or additional text should be included
- The answers should maintain consistent formatting with the correct answer

Example format: { "wrong_answers": ["opt1", "opt2", "opt3"] }

Question: {question}
Correct Answer: {answer}
Generate three wrong answers: """

Table 4: Performance metrics of LLMs on **SimpleQA** dataset in the Normal ($\mathcal{N}$) and Distractor ($\mathcal{D}$) settings, including accuracy (correct), non-attempt (na), ECE, and the number of helped ($\mathcal{D}_{helped}$) and harmed ($\mathcal{D}_{harmed}$) instances. Here the LLM judge model is same as the prediction model.

| LLMs | $\mathcal{N}_{correct}$ | $\mathcal{N}_{none}$ | $\mathcal{N}_{ECE}$ | $\mathcal{D}_{correct}$ | $\mathcal{D}_{none}$ | $\mathcal{D}_{ECE}$ | $\mathcal{D}_{helped}$ | $\mathcal{D}_{harmed}$ |
|---|---|---|---|---|---|---|---|---|
| GPT-4o-mini ◉ | 8.46% | 6.80% | 0.750 | 47.43% | 0.02% | 0.320 | 1644 (93.78%) | 109 |
| GPT-4-turbo ◉ | 20.99% | 7.14% | 0.616 | 65.33% | 0.02% | 0.165 | 1821 (95.44%) | 87 |
| GPT-4o ◉ | 36.75% | 8.16% | 0.437 | 73.48% | 0% | 0.037 | 1507 (91.22%) | 145 |
| LLaMA3.1-8b-instant ∞ | 8.24% | 19.58% | 0.780 | 44.94% | 0.21% | 0.367 | 1294 (91.45%) | 121 |
| LLaMA 3-8B-8192 ∞ | 9.27% | 24.99% | 0.790 | 45.56% | 2.45% | 0.355 | 1251 (90.46%) | 132 |
| Gemma2-9B-it G | 9.52% | 34.49% | 0.771 | 46.58% | 1.87% | 0.359 | 1060 (88.70%) | 135 |

## B  Same LLM judge as Prediction LLM

We employ the same LLM as both the judge and the base model responsible for predicting answers to the questions. The performance metrics are presented in Table 4. Upon manually inspecting instances from smaller LLMs, we observe that the LLM judge occasionally misclassifies responses and refrains from assigning NOT_ATTEMPTED to certain data points. This can be seen by comparing $\mathcal{N}_{none}$ in Table2 and 4 for the SimpleQA dataset. To address this issue and ensure consistency, we use GPT-4o-mini as the LLM judge across all models. We also create reliability diagrams of pipeline where LLM-judge was different and the graphs are shown in Figure 5.

To validate the reliability of GPT-4o-mini as an LLM judge, we conducted a small-scale human evaluation. We sampled 100 responses and had three human annotators independently classify them as CORRECT, INCORRECT, or NOT_ATTEMPTED. The inter-annotator agreement, measured using Cohen's kappa, was 0.82 for GPT-4o-mini, indicating substantial agreement. This comparison allowed us to measure the extent of biases introduced by automated evaluation and confirm that LLM judges generally aligned with human judgments, though minor inconsistencies were observed in ambiguous cases.

## C  Reliability Diagrams of Selected Models on FaVIQ and TriviaQA dataset

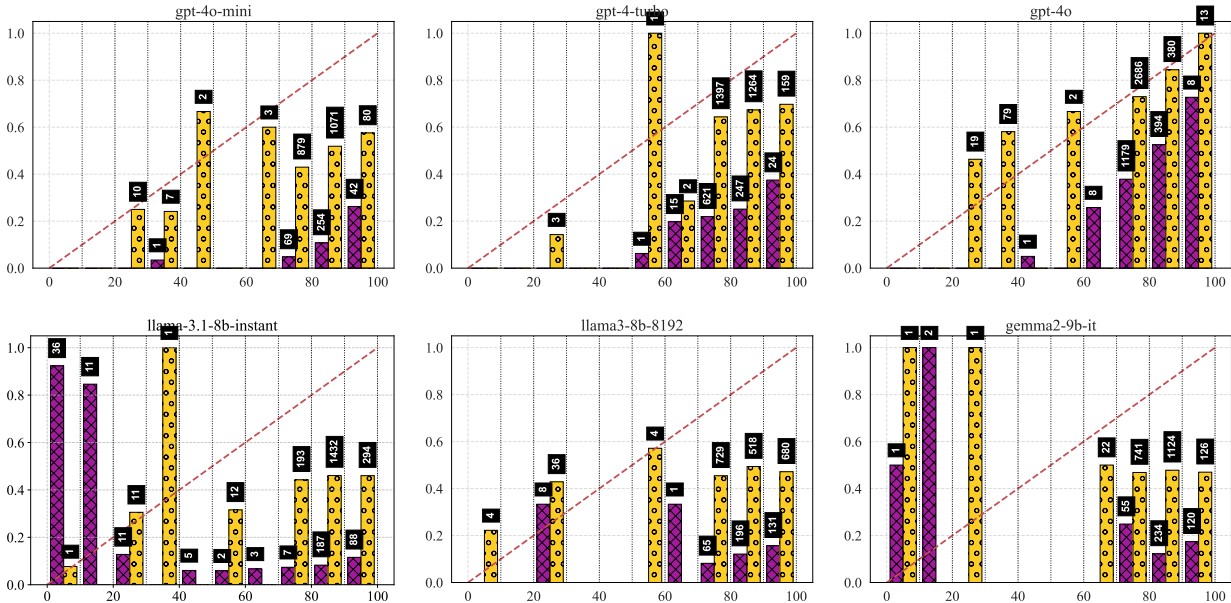

Figure 5: Reliability diagrams (RDs) on SimpleQA dataset showing calibration performance in $\mathcal{N}$ (•) and $\mathcal{D}$ (•) settings. The numbers on top of bars represent the number of correctly predicted instances (y-axis: actual accuracy, x-axis: predicted confidence). Here the LLM judge model is same as the prediction model.

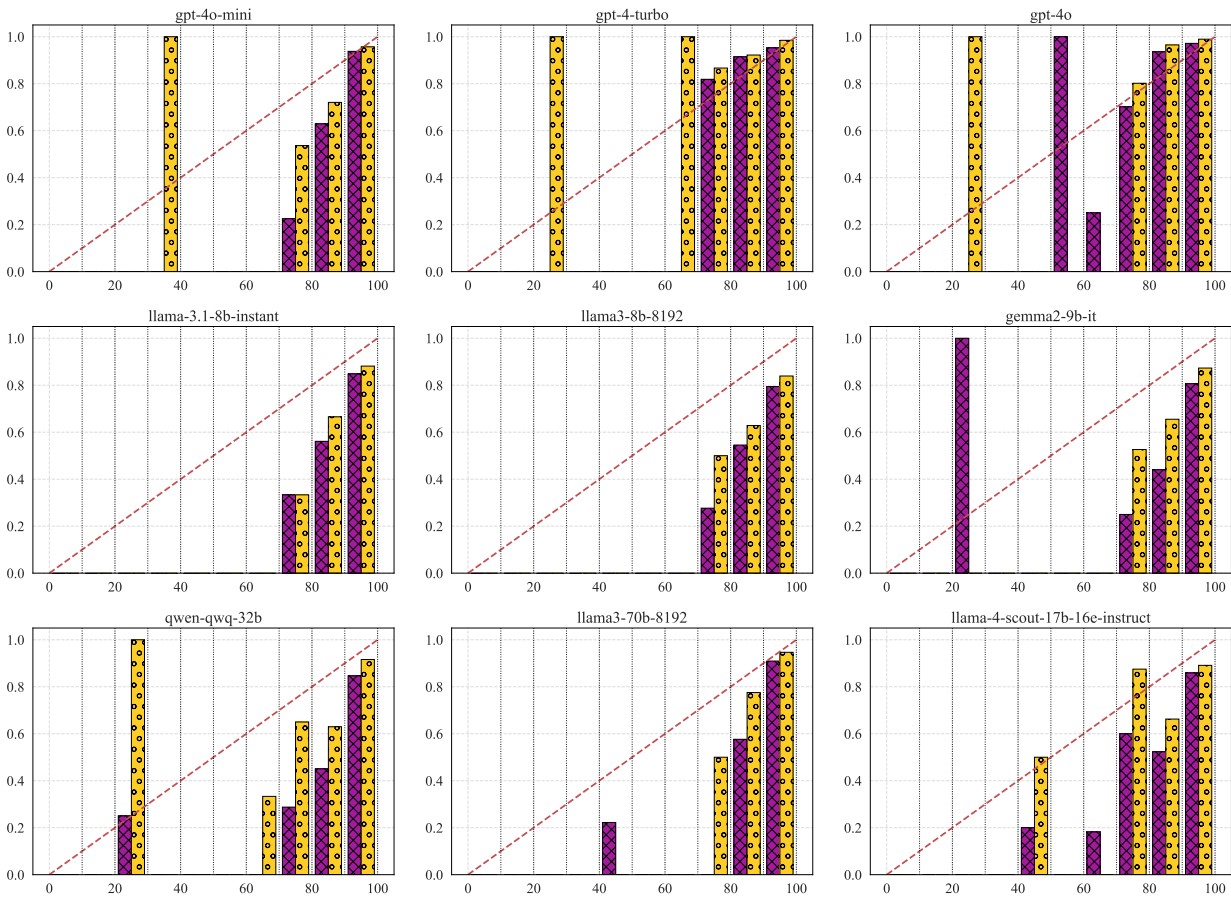

Figure 6: Reliability diagrams (RDs) showing calibration performance in $\mathcal{N}$ (•) and $\mathcal{D}$ (•) settings on the TriviaQA dataset. (y-axis: actual accuracy, x-axis: predicted confidence).

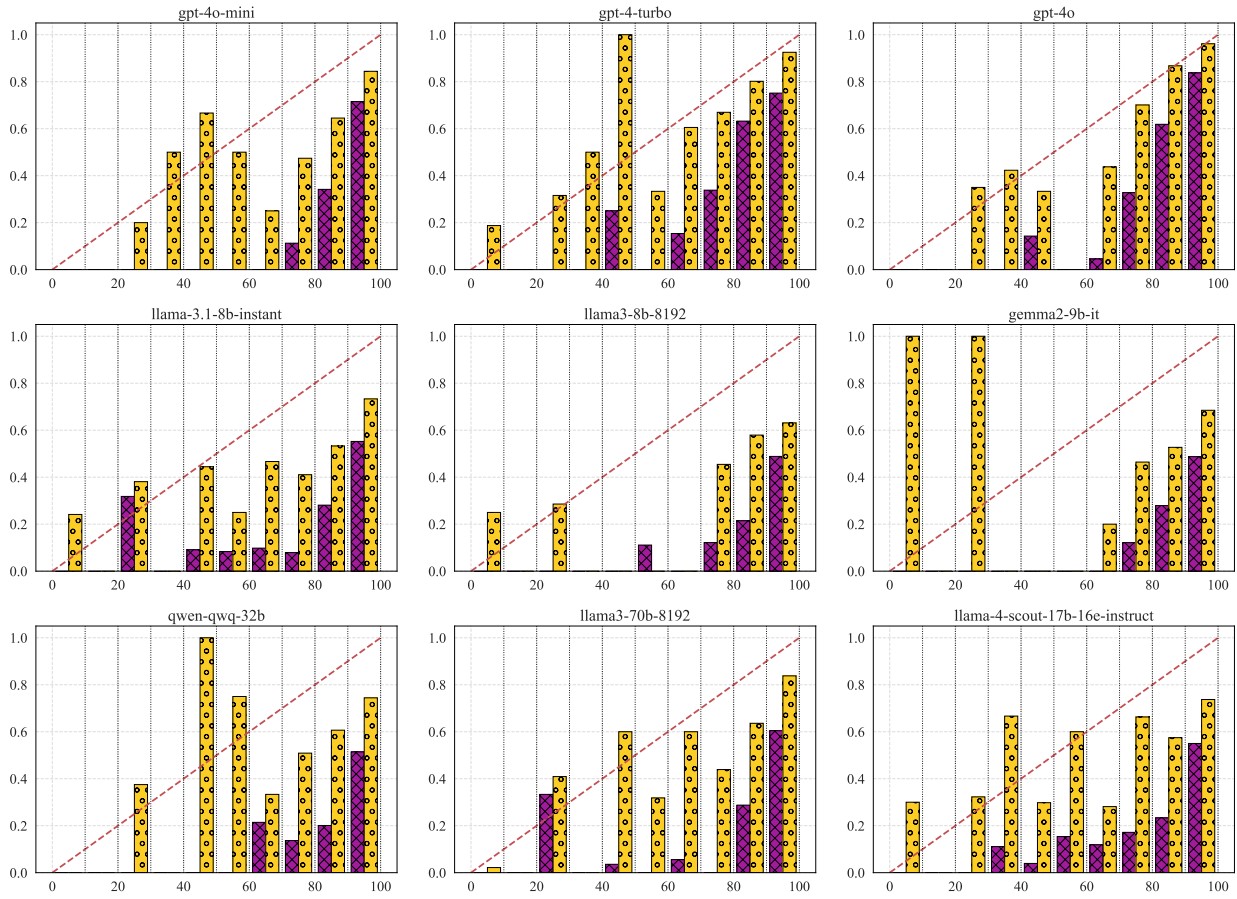

Figure 7: Reliability diagrams (RDs) showing calibration performance in $\mathcal{N}$ (•) and $\mathcal{D}$ (•) settings on the FaVIQ dataset. (y-axis: actual accuracy, x-axis: predicted confidence)

