# OpenReview forum: "Mind the Confidence Gap: Overconfidence, Calibration, and Distractor Effects in Large Language Models"
_TMLR — Accepted by TMLR_

### Review · Reviewer_Aqgr · 2025-09-10

**Summary Of Contributions:**

This paper presents a comprehensive empirical study on the calibration of LLMs. The authors investigate the pervasive issue of overconfidence by evaluating nine different LLMs across three factual question-answering (QA) datasets. The core contribution is the systematic comparison between a standard free-generation setting and a novel "distractor-augmented" setting, where models are prompted with the correct answer alongside several plausible, synthetically generated incorrect options.

Strengths:
- Comprehensive Scope: The evaluation is extensive, covering a wide range of modern LLMs that vary in size, architecture (Dense vs. MoE), and fine-tuning methods (SFT vs. RLHF).

- Novel Methodology: The introduction of the distractor-augmented setting is a key strength. This approach effectively simulates real-world scenarios like multiple-choice questions or retrieval-augmented generation.

- Detailed and Insightful Analysis: The paper goes beyond simple accuracy metrics to provide a fine-grained analysis of calibration, including Expected Calibration Error (ECE), and dissects performance by question type (Person, Place, Date, etc.).

Weaknesses:
- Possibility of Methodological Bias: Both the distractors and the final judgments of correctness were generated by a single LLM (GPT-4o-mini). This introduces a risk of systemic bias, as the distractors may be inherently easier for models from the same family, and the judge may favor certain types of answers.

**Audience:**

Yes

**Audience Explanation:**

The topic of LLM calibration is of critical importance as these models are increasingly deployed in high-stakes applications. This work provides comprehensive and systematic studies on the topic. It moves beyond simply measuring calibration to actively proposing and testing a method (distractor-augmentation) to mitigate miscalibration. The nuanced findings—such as the counterintuitive degradation of calibration in large models on easy tasks and the identification of specific question types prone to miscalibration—are novel and will be of significant interest to researchers or engineers working on LLM safety, interpretability, and robustness.

**Broader Impact Concerns:**

The work does not raise significant ethical concerns. Its primary goal is to improve the reliability and trustworthiness of LLMs, which is a positive contribution to AI safety.

**Claims And Evidence:**

Yes

**Claims Explanation:**

The claims made in the paper are well-supported by extensive empirical evidence. The authors conduct a large-scale, systematic evaluation across nine different models and three datasets, which provides significant weight to their conclusions. The core claims are all substantiated by the quantitative results presented in the tables and figures.

For instance, Table 2 clearly shows the dramatic improvements in accuracy and reductions in ECE when moving from the Normal (N) to the Distractor (D) setting. Figure 3 effectively visualizes the trade-offs, illustrating that while smaller models see massive accuracy gains, their calibration improvement is less pronounced compared to larger models. Figure 4 provides further convincing evidence for the claim that certain categories are inherently more challenging and benefit more from the structured-choice format.

**Requested Changes:**

- Address Potential Judge/Distractor Bias (critical): The reliance on a single model (GPT-4o-mini) for both generating distractors and judging correctness is a significant methodological concern. I strongly recommend the authors expand their analysis of this potential bias. The authors can use a more diverse set of models to generate distractors and/or use a holdout LLM family as the judge.

- Explore Different Distractor Types (strengthening): The current distractors are designed to be "plausible but incorrect." It would strengthen the paper to include an experiment or at least a detailed discussion on how different types of distractors (e.g. common misconceptions or adversarial examples) might affect calibration.

---

> ### Author Response · Authors · 2025-10-02
>
> RC1: We agree that using a single LLM (GPT-4o-mini) as both a distractor generator and judge can introduce systemic bias. We chose a fixed (generator, judge) pair to ensure comparability across nine models (uniform rubric, consistent formatting and criteria). To reduce subjectivity in the current setup, we (i) matched distractor type/format to gold answers, (ii) enforced overlap and mutual-distinctness constraints, and (iii) performed human spot-checks with disagreement resolution. Nevertheless, we will explicitly state in Limitations that results are conditional on this specific (generator, judge) pair, and that judge-family bias is possible. We will also release prompts, distractor sets, and the judging rubric so others can re-score with alternate generators/judges. A cross-family sweep is valuable future work; however, to preserve apples-to-apples comparability and given compute constraints, we keep the fixed setup in this version and temper claims accordingly.
>
> RC2: Our present distractors target the “plausible but incorrect” regime to mimic realistic candidate shortlists (MCQ/RAG triage). We agree that calibration may vary with distractor type (e.g., common misconceptions, near-miss entity swaps, adversarial phrasings, numeric perturbations). In the revision, we will add a concise discussion outlining a taxonomy of distractor families, illustrate each with concrete examples from our released set, and clarify expected calibration impacts (e.g., misconceptions likely raise confidence on wrong answers; near-misses stress discrimination without necessarily inflating confidence). We will explicitly scope this study to the current “plausible-incorrect” design and flag broader distractor families as future work, to avoid over-claiming.

---

### Review · Reviewer_hv9U · 2025-09-23

**Summary Of Contributions:**

This paper conducts a study to figure out when and why LLMs act overconfident. The authors ask the LLM to predict answer and confidence score and measure the confidence score calibration error using ECE metrics. Their main contributions are
1. Testing LLMs with Structured Distractors: The key idea is to compare how an LLM answers a question in two ways: (1) just asking it directly (free-generation) versus (2) giving it a multiple-choice question with one correct answer and three plausible-but-wrong answers (called "distractors"). Then study the affect of distractors on accuracy and confidence score calibration.
2. Detailed Breakdown of Failures and Model Capability Analysis: They ran this experiment on nine different popular LLMs (including various GPT, LLAMA, and other models) across three different question-answering datasets to see if the results were consistent. Instead of just a single "pass/fail" score, they dug into the details. They analyzed what makes models more or less overconfident, looking at factors like:
(a) Model Size: Do bigger models (like GPT-4o) behave differently than smaller ones (like LLAMA-8B)?
(b) Training Method: Does the "RLHF" tuning that makes models good at chatting also make them more or less overconfident?
(c) Question Type: Are models more overconfident about "people" than they are about "dates" or "places"?

**Audience:**

Yes

**Audience Explanation:**

[Borderline Yes]
* The central theme of the paper is the "misalignment between predicted confidence and true correctness" (miscalibration). This is one of the most significant barriers to deploying LLMs in critical decision-making applications. The audience, which includes researchers focused on AI safety, robustness, and reliability, is deeply invested in understanding and solving this exact problem. Therefore, it would be interesting topic to them.
* The paper provides more than just confirmatory results. The discovery that large, advanced models (like GPT-4-turbo and GPT-4o) can "paradoxically suffer increased miscalibration on easier queries"  is a novel, surprising, and important finding. This kind of non-obvious finding and analysis would be interesting to the audience.
* The analytical rigor in this paper is very good. The authors haven't only provided top line scores but also analyzed the score at different angles - how scale, tuning regime, and architecture independently influence model calibration.

The reason why I have borderline yes vote, is due to the following concerns
* The study is performed exclusively on three factual Question-Answering datasets. The findings may not apply to other common LLM tasks, such as creative writing, summarization, mathematical reasoning, or code generation.
* The distractors format is fixed: one correct answer plus three distractors. This doesn't fully capture real-world scenarios (like Retrieval-Augmented Generation) where a model might receive five retrieved chunks, none of which are fully correct, or all of which are partially correct. The findings may not hold in a "noisy" environment with a different number of distractors or no guaranteed correct answer.

**Claims And Evidence:**

Yes

**Claims Explanation:**

[Borderline Yes]
* This paper claims that LLMs are generally overconfident and adding structured distractors as options mitigate this problem.
* The authors have studied the LLMs overconfidence problem by asking LLM directly about its confidence in the generated answer. Standard datasets (SimpleQA, FaVIQ, TriviaQA) were used to measure the quality of answers and confidence score estimations. The prompts for answer generation, structured distractor generation and LLM Juding are provided. The results from judging and structured distractor generation prompting are said to be manually verified by 3 reviews which I believe to be true.
* The authors have not only provided the metrics but also provided rationale and plausible reasoning for the results and unexpected results.

The reason why I have borderline yes vote, is due to the following concerns
* Confidence score generation via LLM prompting isn't the only way of generating confidence score through LLMs. Model's internal logits can also be utilized to generate confidence score. Similarly majority voting approach can be used as well. This paper doesn't provide any details of other confidence score generation algorithms and affect of distractors on the confidence score calibration.
* The LLM was asked to provide confidence score in 3 wide bins. In my opinion, criteria for more granular confidence score estimation should have been given.
* The entire distractor-augmented setting relies on the quality of the three incorrect answers. These distractors were generated by GPT-4o-mini. The authors state they were carefully designed and manually inspected, but the "plausibility" of these distractors is a critical, subjective variable. If the distractors are too simplistic or "obviously wrong," the task becomes trivial and the calibration findings are skewed. The experiment's validity is thus dependent on the quality of an AI-generated artifact.
* There is a risk of systemic bias, where the judge (GPT-4o-mini) may favor the answer style or factual knowledge of its own model family (the GPT series), potentially inflating their performance scores relative to other models like LLAMA or Qwen.

**Requested Changes:**

**Issue**: The paper measures confidence by prompting the LLM for a 0-100 score. This is a valid method, but it is not the only one. Other methods, such as using the model's internal logits or employing a majority voting approach, are not explored or justified.
**Proposed Adjustment**: The authors must add a section (in Methodology or Limitations) that explicitly justifies their choice of using elicited confidence. This discussion should acknowledge the existence of alternative methods (like logit-based confidence) and critically evaluate how this choice might impact the findings.

**Issue**: The experimental design relies heavily on a single model (GPT-4o-mini) for two key functions:
Distractor Generation: The "plausibility" of the AI-generated distractors is a critical, subjective variable that underpins the entire distractor-augmented experiment.
Answer Adjudication: The same model is used as the judge to determine correctness, which risks systemic bias (e.g., favoring the style or factual knowledge of its own model family).
**Proposed Adjustment**: While the authors do perform some validation, they should explicitly discuss the risk of bias from this "AI-judging-AI-generated-content" loop in their Limitations section. Acknowledging that the results are contingent on this specific generator/judge model's behavior is necessary for full transparency.
One option to mitigate this problem would be to generate distractors and evaluate the scores using other LLMs as well. Then provide provide a wholistic evaluation analysis.

---

> ### Author Response · Authors · 2025-10-02
>
> Issue 1: We appreciate the reviewer’s concern regarding the choice of elicited confidence scores over logit-based or majority-vote approaches. Our decision was intentional, and we will make this explicit in the revised manuscript. Specifically:
>
> 1. Logit-based confidence measures often reflect surface-level fluency rather than true task-level belief. Lin et al. [1] show that eliciting verbalized probabilities directly from LLMs can provide a more faithful signal of what the model believes than relying solely on log-probabilities, which can be distorted by tokenization artifacts and synonym choice.
> 2. Tian et al. [2] demonstrate that RLHF-tuned models are often worse calibrated when judged purely by token-level probabilities, but verbalized/elicited confidence mitigates this effect and even yields lower ECE. This aligns with our focus on elicited confidence as a robust and user-facing measure in RLHF-aligned systems.
> 3. Mielke et al. [3] highlight that elicited verbal confidence aligns a model’s language with its underlying correctness likelihood, reducing systematic overconfidence and providing more transparent signals for end users. This directly supports our methodological choice to analyze overconfidence through elicited scores.
> 4. Not all models in our study expose logits or support extensive sampling required for self-consistency methods. Elicited confidence, however, is available across closed and open weight systems, making it the only uniformly applicable signal for a multi-model comparison.
>
> We will clarify that our goal was not to exhaust all possible confidence-estimation methods, but to analyze elicited confidence as a first-class signal, an approach supported by recent literature and practical constraints.
>
> [1] https://arxiv.org/pdf/2205.14334
> [2] https://arxiv.org/pdf/2305.14975
> [3] https://arxiv.org/pdf/2012.14983
>
> Issue 2: Thank you for flagging the potential bias in using one model both to generate distractors and to adjudicate answers. Our choice was intentional for consistency and comparability: a fixed generator and a fixed judge avoid cross-model drift in style, formatting, and rubric, which can otherwise confound multi-model comparisons. To reduce subjectivity in distractor quality, we enforced type and format matching to the gold answer, rejected near-duplicates, and applied simple plausibility checks (e.g., length and overlap constraints), followed by human spot-checks by three reviewers with disagreement resolution. Using a single judge for all systems also avoids per-model rubric shifts that could inadvertently favor different families. That said, we agree the “AI-generates / AI-judges” loop can introduce systemic bias. In the revision, we will explicitly acknowledge this limitation and make clear that our findings are conditional on the (generator, judge) pair we fixed.

---

> > ### Comment · Reviewer_hv9U · 2025-11-06
> > **Response to the authors**
> >
> > I thank the authors for their detailed response to my previous concerns.
> >
> > Original Issue #1: Confidence score generation via LLM prompting isn't the only way of generating confidence score through LLMs
> > Response to Authors: I understand the rationale for prioritizing elicited confidence scores over logit-based or majority-vote approaches, particularly given the practical limitations of those alternatives in heterogeneous model environments. However, without a direct comparison to other confidence estimation methods, it remains difficult to fully isolate the specific impact of the distractor-based confidence generation method. Acknowledging this limitation and explicitly including the rationale provided in your rebuttal in the revised paper will help readers better understand the methodological context.
> > Conclusion: Mitigated.
> >
> > Original Issue #2: Potential bias in using one model both to generate distractors and to adjudicate answers
> > Response to Authors: While I accept that using the same LLM (GPT-4o-mini) for both distractor generation and judging helps avoid cross-model drift in style and rubrics, the concern regarding systemic bias is not fully mitigated. Although you employed heuristics and human review of samples, these measures do not entirely eliminate the potential for the judge to favor its own generation patterns. A more robust, albeit cost-intensive, approach would involve validating trends using alternating pairs of different LLMs for generation and judging. However, explicitly detailing this limitation and making it clear to readers that findings are conditional on this specific generator/judge pair in the next revision is an acceptable compromise for transparency.
> > Conclusion: Not fully mitigated, but acceptable if the limitation is explicitly stated in the revision.

---

> > > ### Author Response · Authors · 2025-11-10
> > >
> > > Response 1: Thank you for the clarifying feedback. In the revision we will (i) add a short paragraph in Methodology justifying our use of elicited confidence, (ii) explicitly mention alternative estimators (logit-based, etc) and the API/black-box constraints that prevented a uniform comparison, and (iii) state in Limitations that our findings are conditional on elicited confidence and may vary under other estimators.
> > >
> > > Response 2: We agree that using the same LLM (GPT-4o-mini) for both distractor generation and judging leaves a channel for systemic bias, since the judge may be better aligned with patterns it helped create. In our pipeline, we sampled distractors with a higher temperature (=1) to increase plausibility/diversity, and we judged at temperature 0 to make scoring deterministic and comparable across all 9 systems. This choice reduces variance and filters obviously weak distractors, but, as you note, it does not remove the model-family dependency. In the revision, we will state this explicitly in the Limitations section and clarify that our findings should be read as conditional on this fixed GPT-4o-mini (generator, judge) pair. We will also mention your suggested, more robust (but costlier) extension of replaying the study with alternating generator/judge LLMs to confirm that the trends persist.

---

> > > > ### Comment · Reviewer_hv9U · 2025-11-11
> > > > **Response to the authors**
> > > >
> > > > I thank the authors for their constructive engagement and clear plan for revision.
> > > >
> > > > Regarding Issue 1 (Confidence Estimation): The proposed updates to the Methodology and Limitations sections, specifically justifying the elicited confidence choice, and noting the constraints that prevented their uniform application fully address my concern.
> > > >
> > > > Regarding Issue 2 (Generator/Judge Bias): Stating that the findings are conditional on the fixed GPT-4o-mini pair, while also mentioning the suggested alternating-LLM validation method as a potential avenue for future confirmation, is a satisfactory resolution for this submission.
> > > >
> > > > I look forward to seeing these clarifications in the revision.

---

### Review · Reviewer_raGB · 2025-10-06

**Summary Of Contributions:**

This paper conducts a large-scale empirical study on the calibration of large language models (LLMs). The authors analyze different model families, sizes, and architectures across three factual question answering (QA) datasets of varying difficulty, and ultimately offer recommendations for deploying more reliable LLMs.

**Audience:**

Yes

**Audience Explanation:**

Yes, LLM calibration is a relatively important issue and will attract some people’s interest.

**Broader Impact Concerns:**

No.

**Claims And Evidence:**

No

**Claims Explanation:**

1. The paper does not compare against any other calibration methods. Without such a comparison, we can’t tell whether the proposed distractor-based prompt engineering is truly a superior calibration technique, or merely a more complex and costly way to achieve what a simpler method could do.

2. The study obtain confidence by explicitly asking the model to provide a “confidence_score” between 0 and 100. Which is better: this verbalized confidence approach, or a method based on internal token-level log probabilities of the model’s outputs? If model calibration is poor, is it because the model is overconfident about its knowledge, or because it struggles with the secondary task of mapping its internal uncertainty onto the numeric value requested in the prompt?

**Requested Changes:**

Please address the concerns raised in my explanation regarding the criterion“claims made in the submission supported by accurate, convincing and clear evidence”.

---

> ### Author Response · Authors · 2025-10-15
>
> Response 1: Our contribution is primarily an evaluation paradigm showing how structured distractors reshape both accuracy and calibration across models and datasets, not a claim that distractor prompting strictly dominates post-hoc calibration. We will make this explicit up front and soften any language that might imply superiority. Our current text already frames post-hoc calibration as underexplored for modern LLMs and motivates studying distractors as a missing piece. We will surface this context earlier and more clearly. We measure calibration via ECE with 0.1 bins and track how distractors help or harm instances (D_helped/D_harmed) in addition to accuracy, these will remain our core readouts.
>
> Response 2: We appreciate the reviewer’s concern regarding the choice of elicited confidence scores (0-100) over logit-based or majority-vote approaches. Our decision was intentional, and we will make this explicit in the revised manuscript. Specifically:
>
> 1. Logit-based confidence measures often reflect surface-level fluency rather than true task-level belief. Lin et al. [1] show that eliciting verbalized probabilities directly from LLMs can provide a more faithful signal of what the model believes than relying solely on log-probabilities, which can be distorted by tokenization artifacts and synonym choice.
> 2. Tian et al. [2] demonstrate that RLHF-tuned models are often worse calibrated when judged purely by token-level probabilities, but verbalized/elicited confidence mitigates this effect and even yields lower ECE. This aligns with our focus on elicited confidence as a robust and user-facing measure in RLHF-aligned systems.
> 3. Mielke et al. [3] highlight that elicited verbal confidence aligns a model’s language with its underlying correctness likelihood, reducing systematic overconfidence and providing more transparent signals for end users. This directly supports our methodological choice to analyze overconfidence through elicited scores.
> 4. Not all models in our study expose logits or support extensive sampling required for self-consistency methods. Elicited confidence, however, is available across closed and open weight systems, making it the only uniformly applicable signal for a multi-model comparison.
>
> We will clarify that our goal was not to exhaust all possible confidence-estimation methods, but to analyze elicited confidence as a first-class signal, an approach supported by recent literature and practical constraints.
>
> [1] https://arxiv.org/pdf/2205.14334
> [2] https://arxiv.org/pdf/2305.14975
> [3] https://arxiv.org/pdf/2012.14983

---

### Decision · Action_Editor_62JM · 2025-11-26

**Recommendation:** Accept with minor revision

**Audience:**

Yes

**Audience Explanation:**

The problem of calibration and overcondfidence is a very important and widespread issue for current LLMs and is of interest for a huge audience. Moreover, this work, although with constraints and bias given reliance on a single model, can be used as the first step to more investigation in this area.

**Claims And Evidence:**

Yes

**Claims Explanation:**

The paper focuses on the well spread problem of overconfidence of LLMs, especially for QA tasks. They look into 9 LLMs and 3 QA tasks. The authors addressed reviewers concerns well and the remaining requirement is to clearly adding the following details to the paper:

Methodological Rigor (Confidence Estimation): The authors provided a sound justification for relying on elicited confidence scores. They cited relevant literature regarding their efficacy in RLHF models and noted the practical constraints of uniform logit access across standard APIs. Their commitment to explicitly detailing this rationale and its established trade-offs in the revised Methodology mitigates my concerns.

Addressing Potential Bias: While the reliance on a single model (GPT-4o-mini) for both distractor generation and answer adjudication introduces potential systemic bias, the authors have agreed to explicitly frame their findings as conditional on this specific pipeline in the Limitations section. They also agreed to document the suggested "alternating LLM" validation approach as a necessary avenue for future work to confirm generalizability.

Moreover, this work, although with constraints and bias given reliance on a single model, can be used as the first step to more investigation in this area.

---

> ### Author Response · Authors · 2025-12-02
>
> Thank you to the reviewers and the Action Editor for the thoughtful and constructive feedback. We have carefully incorporated all recommendations, and the camera-ready version of the paper has now been uploaded. We appreciate your time and effort in helping strengthen this work.